 **eLIFE**

# Lys63-linked ubiquitin chain adopts multiple conformational states for specific target recognition

**Zhu Liu[1,2], Zhou Gong[1], Wen-Xue Jiang[1], Ju Yang[1], Wen-Kai Zhu[1], Da-Chuan Guo[1], Wei-Ping Zhang[2], Mai-Li Liu[1]\*, Chun Tang[1]\***

[1]CAS Key Laboratory of Magnetic Resonance in Biological Systems, State Key Laboratory of Magnetic Resonance and Atomic Molecular Physics, Wuhan Institute of Physics and Mathematics of the Chinese Academy of Sciences, Wuhan, China; [2]Department of Pharmacology and Institute of Neuroscience, Zhejiang University School of Medicine, Hangzhou, China

**Abstract** A polyubiquitin comprises multiple covalently linked ubiquitins and recognizes myriad targets. Free or bound to ligands, polyubiquitins are found in different arrangements of ubiquitin subunits. To understand the structural basis for polyubiquitin quaternary plasticity and to explore the target recognition mechanism, we characterize the conformational space of Lys63-linked diubiquitin (K63-Ub$_2$). Refining against inter-subunit paramagnetic NMR data, we show that free K63-Ub$_2$ exists as a dynamic ensemble comprising multiple closed and open quaternary states. The quaternary dynamics enables K63-Ub$_2$ to be specifically recognized in a variety of signaling pathways. When binding to a target protein, one of the preexisting quaternary states is selected and stabilized. A point mutation that shifts the equilibrium between the different states modulates the binding affinities towards K63-Ub$_2$ ligands. This conformational selection mechanism at the quaternary level may be used by polyubiquitins of different lengths and linkages for target recognition.

**\*For correspondence:** tanglab@ wipm.ac.cn (CT); ml.liu@wipm.ac. cn (M-LL)

**Competing interests:** The authors declare that no competing interests exist.

## Introduction

Ubiquitin is a 76-residue signaling protein found ubiquitously in cells. Multiple ubiquitins are covalently linked to form a polyubiquitin, which can then be attached to a substrate protein. The process is known as ubiquitination, a post-translational modification of the substrate protein. Three classes of enzymes catalyze ubiquitination: ubiquitin-activation enzyme (E1), ubiquitin-conjugation enzymes (E2), and ubiquitin-protein ligases (E3). E2 and E3 dictate ubiquitin linkage and substrate specificities. Additionally, deubiquitinases (DUBs) are responsible for specifically erasing ubiquitin signals from a substrate protein (*Clague et al., 2012*).

Catalyzed by a linkage-specific E2, two or more ubiquitins are linked up via an isopeptide bond between the carboxylate at the C-terminus of one ubiquitin (referred to as the distal unit) and the ε-amine of a lysine residue (Lys6, Lys11, Lys27, Lys29, Lys33, Lys48, or Lys63) or the α-amine at the N-terminus of another ubiquitin (referred as the proximal unit). Lys48-linked polyubiquitin is the most abundant linkage in cells, and signals substrate proteins for proteasomal degradation (*Kravtsova-Ivantsiv et al., 2013*; *Lu et al., 2015*). Other types of linkages mostly perform non-degradative functions (*Xu et al., 2009*; *Kulathu and Komander, 2012*). Lys63-linked polyubiquitin is another common linkage, and has been found to be involved in DNA damage response (*Hoege et al., 2002*), multivesicular body mediated protein sorting (*MacDonald et al., 2012*), NF-κB signaling (*Chen and Chen, 2013*; *Zinngrebe et al., 2014*), and oxidative stress response (*Silva et al., 2015*). Lys63-linked polyubiquitin also exists as an unanchored form without being attached to a substrate protein, and

**eLife digest** Proteins can be tagged with other molecules that indicate what the cell should do with that protein. For example, proteins tagged with a small protein called ubiquitin—which is linked to other ubiquitin molecules to form 'polyubiquitin'—may be destroyed or relocated within a cell.

Like all proteins, a ubiquitin is made up of chains of amino acids. Specific amino acids form the linkages between individual ubiquitins to form a polyubiquitin, and the nature of these linkages influences the effect that a polyubiquitin has on the tagged protein. One linkage involves a lysine amino acid at position 63 (known as Lys63). This linkage is found in the polyubiquitin that is involved in repairing damaged proteins and relocating target proteins to a part of the cell where they are utilized for immune response. To perform these different roles, the polyubiquitin must be able to distinguish between a variety of target proteins.

The shape that a protein takes on determines how it works, and most proteins constantly and rapidly switch between different shapes. Previous work suggested that the Lys63-linked polyubiquitin could only take on an elongated 'open' structure by itself. It was not clear whether the protein could take on a compact 'closed' structure without first binding to a target protein.

Liu et al. used a technique known as nuclear magnetic resonance (NMR) to explore the high-resolution structures of the Lys63-linked ubiquitin chain when they are not bound to other proteins. The results showed that a large percentage of the protein was in a closed state, and that there were at least one open shape and two kinds of closed shapes.

Liu et al. suggest that the shape of the unbound Lys63-linked ubiquitin chain determines what other proteins can be bound, and that the binding stabilizes the shape of the ubiquitin. This mechanism of binding is known as conformational selection. Further work is required to analyze whether other polyubiquitin chains recognize their partners in a similar manner.

the unanchored form may serve as a scaffold for recruiting proteins in the signaling pathways of innate immunity and protein aggregate removal (*Zeng et al., 2010*; *Hao et al., 2013*).

How does Lys63-linked polyubiquitin perform different functions? To do so, Lys63-linked polyubiquitin has to specifically recognize multiple target proteins. Structural studies of Lys63-linked diubiquitin (K63-Ub$_2$) have indicated that K63-Ub$_2$ mostly adopts open extended conformations in the absence of a ligand and in complex with many target proteins (*Datta et al., 2009*; *Komander et al., 2009*; *Sato et al., 2009a*; *Weeks et al., 2009*; *Sekiyama et al., 2012*). It has been proposed that such an open extended structure differentiates K63-Ub$_2$ from Lys48-linked polyubiquitin that predominantly exists in a closed structure (*Varadan et al., 2004*; *Fushman and Walker, 2010*; *Hirano et al., 2011*). Notwithstanding, K63-Ub$_2$ has been found in closed conformations when complexed with the Npl4 zinc-finger domain (NZF) of TAK1-binding proteins (*Kulathu et al., 2009*; *Sato et al., 2009b*) for the activation of NF-κB signaling pathways. K63-Ub$_2$ is also selectively recognized by the fourth zinc-finger (ZnF4) of A20, a ubiquitin-editing enzyme for the termination of NF-κB signaling (*Wertz, 2014*; *Wertz and Dixit, 2014*). The crystal structure of the complex between ZnF4 and ubiquitin monomer indicated that K63-Ub$_2$ likely exists in a closed conformation when binding to ZnF4 (*Bosanac et al., 2010*). So are the closed-state structures already present for the free K63-Ub$_2$, but have hitherto eluded structural characterization? Or is the closed-state structure induced by a specific ligand?

To recognize a target protein and to perform specific functions, a protein has to fluctuate among a variety of conformational states (*Henzler-Wildman and Kern, 2007*). For a ubiquitin monomer, studies have shown that the protein exists as an ensemble of tertiary conformations which can accommodate different target proteins (*Lange et al., 2008*). As ubiquitin mainly functions as polyubiquitins, a more pertinent question is how does a polyubiquitin fluctuate at the quaternary level and achieve its target recognition specificity?

Nuclear magnetic resonance (NMR) is well suited to characterize protein ensemble structures and to visualize protein dynamics. NMR depiction of dynamic fluctuation for multi-domain proteins such as pre-mRNA splicing factor U2AF and DNA-binding protein CAP, has uncovered conformational selection and equilibrium shift mechanisms for these systems (*Mackereth et al., 2011*; *Tzeng and Kalodimos, 2012*; *Huang et al., 2014*). Among the NMR techniques, paramagnetic relaxation

enhancement (PRE) is exquisitely sensitive to transient and fleeting interactions between proteins (*Sekhar and Kay, 2013*; *Xing et al., 2014*). Owing to dipolar interactions between the paramagnetic center and protein nuclei, PRE is proportional to the inverse sixth power of the distance between the paramagnetic center and protein nuclei, and is ensemble-averaged over all the conformational states sampled (*Clore and Iwahara, 2009*). To explore the target recognition mechanism for K63-Ub$_2$, we used PRE NMR in conjunction with other biophysical methods. We characterized the arrangements between the ubiquitin units, and we determined the ensemble structure for K63-Ub$_2$ in the absence of a ligand. Our findings indicated a conformational selection mechanism at the quaternary level, whereby a target protein selects and stabilizes one of the preexisting conformational states of ligand-free K63-Ub$_2$.

## Results

### Ligand-free K63-Ub$_2$ exists in both open and closed states

We first compared the chemical shift differences between the subunits in K63-Ub$_2$ and ubiquitin monomer. Except for residues near the covalent ubiquitin linkage, the differences in chemical shifts are small (<0.04 ppm; *Figure 1—figure supplement 1*). Residues with relatively large chemical shift differences (>0.01 ppm) can be tentatively mapped, and form rather contiguous surfaces on each subunit (*Figure 1—figure supplement 1*, insets). However, it is unclear whether the perturbations are due to the covalent linkage, or due to non-covalent interactions between the two subunits. We also performed a half-filtered NMR experiment, which failed to reveal any inter-subunit nuclear Overhauser effects (NOEs) between the proximal and distal units of K63-Ub$_2$. Our data are consistent with the previous NMR studies of K63-Ub$_2$ (*Tenno et al., 2004*; *Varadan et al., 2004*); in the latter work, the authors failed to detect cross-saturations between the two subunits. Together, the diamagnetic NMR studies indicated that the closed-state structure of K63-Ub$_2$, if existing, is loosely packed and possibly adopts multiple conformations.

To visualize the quaternary arrangement between the subunits of K63-Ub$_2$, we resorted to PRE NMR. We prepared K63-Ub$_2$ protein with the proximal unit $^{15}$N-labeled and the distal unit unlabeled. A cysteine point mutation was introduced to Asn25 or Lys48 in the distal unit of K63-Ub$_2$. An MTS paramagnetic probe was conjugated at N25C or K48C site. Each of the conjugation sites was designed so that the paramagnetic probe was away from the binding partners of K63-Ub$_2$, which include Rap80 tUIM domain (*Sekiyama et al., 2012*), TAB2 NZF domain (*Kulathu et al., 2009*; *Sato et al., 2009b*), and A20 ZnF4 domain (*Bosanac et al., 2010*), and therefore the conjugation does not interfere with ligand binding (*Figure 1—figure supplement 2*). Indeed, the binding affinities between K63-Ub$_2$ and tUIM or NZF remain unchanged for the paramagnetically tagged K63-Ub$_2$ proteins (*Figure 1—figure supplement 3*). In addition, the paramagnetic NMR spectrum can be overlaid onto the diamagnetic spectrum, except for residues that are completely broadened out due to the PRE effect (*Figure 1—figure supplement 4*). Together, the modifications do not perturb the conformational space of K63-Ub$_2$ or have an effect on K63-Ub$_2$ function.

We previously reported that ubiquitin monomer dimerizes non-covalently with an apparent $K_D$ value of 4.9 ± 0.3 mM (*Liu et al., 2012*). Therefore, the PRE effect could arise both intramolecularly and inter-molecularly. The inter-molecular PREs were measured on an equimolar mixture of K63-Ub$_2$ (each at 500 μM), with paramagnetic tagging and isotope labeling on different subunits in separate proteins. With a paramagnetic probe conjugated at either N25C or K48C site and using the inter-molecular data for reference, our measurements revealed large intra-molecular inter-subunit PREs for many residues in the proximal unit (*Figure 1A,B*). At relatively low protein concentration (50 μM), the inter-molecular contribution to the overall PRE is negligible. We found that the relative decreases in peak intensities between the paramagnetic and diamagnetic spectra recorded at 50 μM are highly correlated with the relative decreases between the paramagnetic and inter-molecular spectra recorded at 500 μM (*Figure 1—figure supplement 5*). This corroborates the PRE measurement at the higher concentration. In addition, when a different paramagnetic probe, EDTA-Mn$^{2+}$, was conjugated at N25C, the PRE profile is similar to that using the MTS probe (*Figure 1—figure supplement 6*). Thus, the intra-molecular inter-subunit PREs are independent of the paramagnetic probe used, and reveal intrinsic structural features of ligand-free K63-Ub$_2$.

K63-Ub$_2$ has been generally considered to only exist in the open state for both ligand-free and many ligand-bound forms. For the known open-state structures of K63-Ub$_2$ (*Sato et al., 2008*;

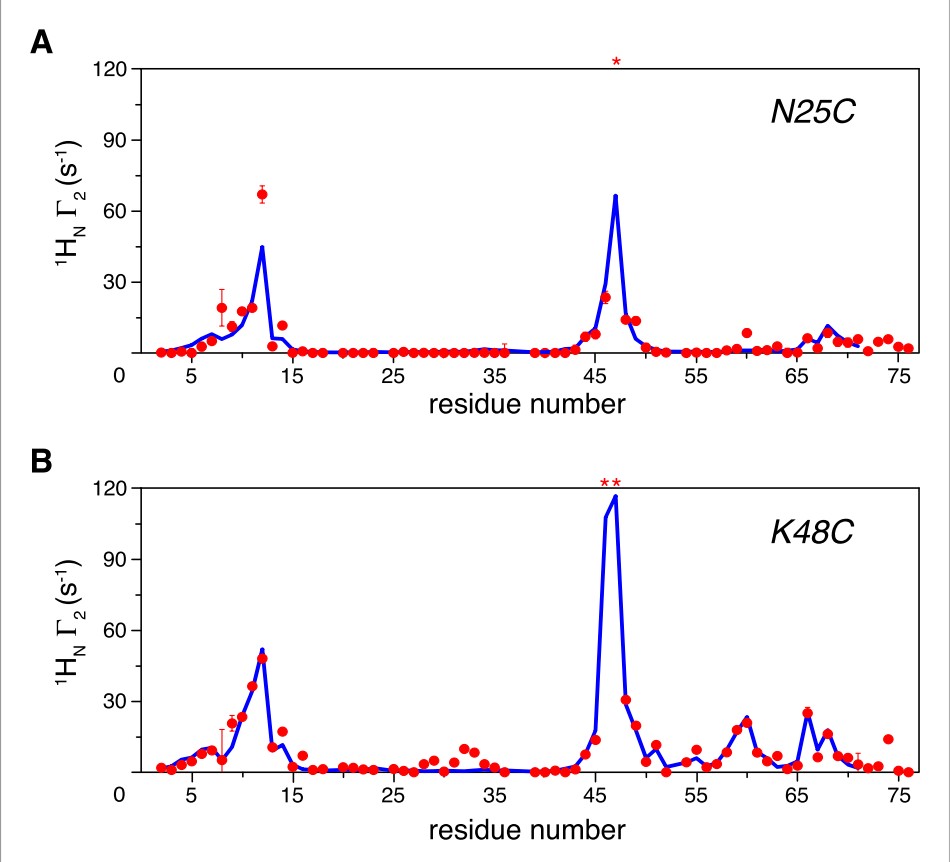

**Figure 1**. Intra-molecular inter-subunit paramagnetic relaxation enhancements (PREs) measured for ligand-free K63-Ub$_2$. With the paramagnetic probe conjugated at (**A**) N25C or (**B**) K48C in the distal unit, the PRE $^1$H $\Gamma_2$ rates were measured for amide protons of the $^{15}$N-labeled proximal unit. The red spheres indicate the observed PREs. The error bar indicates 1 SD in the PRE measurement. The blue lines are the back-calculated PRE values for residues 1–71. Residues that are completely broadened out are denoted with asterisks at the top.

The following figure supplements are available for figure 1:

**Figure supplement 1**. NMR chemical shift differences between ubiquitin monomer and K63-Ub$_2$.

**Figure supplement 2**. Illustration of cysteine point mutation and conjugation of an MTS paramagnetic probe to K63-Ub$_2$, which is complexed with (**A**) Rap80 tUIM, (**B**) TAB2 NZF, or (**C**) A20 ZnF4.

**Figure supplement 3**. Isothermal calorimetry (ITC) measurements for the binding affinities (**A**–**C**) between K63-Ub$_2$ and tUIM, and (**D**–**F**) between K63-Ub$_2$ and NZF.

**Figure supplement 4**. Overlay of 2D NMR spectra for wild type protein and paramagnetically tagged K63-Ub$_2$ proteins at (**A**) N25C site and (**B**) K48C site.

**Figure supplement 5**. Correlations of the paramagnetic effects measured at two different concentrations.

**Figure supplement 6**. Comparison of the intra-molecular inter-subunit paramagnetic relaxation enhancement (PRE) data with an EDTA-Mn$^{2+}$ (red circles) or MTS probe (blue circles) conjugated at N25C site.

**Figure supplement 7**. Comparison between all known structures of K63-Ub$_2$ in the open state.

**Figure supplement 8**. Intra-molecular inter-subunit paramagnetic relaxation enhancements (PREs) arising from K63-Ub$_2$ open state are negligible.

*Datta et al., 2009*; *Komander et al., 2009*; *Sato et al., 2009a*; *Weeks et al., 2009*; *Yoshikawa et al., 2009*; *Sekiyama et al., 2012*), the intra-molecular inter-subunit PREs calculated with an MTS probe attached at either N25C or K48C site are essentially zero for residues in the proximal unit (*Figure 1—figure supplement 7*). Alternatively, an open extended conformation of $K63-Ub_2$ can be simply obtained by restraining the inter-subunit PRE target value to zero for residues in the proximal unit—the resulting conformational space encompasses all known $K63-Ub_2$ structures in the open state (*Figure 1—figure supplement 8*). As such, the large inter-subunit PREs should only arise from the closed state of $K63-Ub_2$, and ligand-free $K63-Ub_2$ should exist in both open and closed states.

The existence of the closed state for ligand-free $K63-Ub_2$ is corroborated by small angle X-ray scattering (SAXS) analysis. The SAXS data collected for $K63-Ub_2$ at higher concentrations display larger particle size than those at lower concentrations, indicative of high-order oligomers for the former (*Figure 2—figure supplement 1*). At lower concentrations, the $D_{max}$ value is smaller, and the data recorded at 1 mM and 500 µM are similar ($D_{max}$ = 67.2 and 65 Å, respectively). The $D_{max}$ values are smaller than the calculated values for all known open-state structures (84.0 ± 3.3 Å). Significantly, the experimental paired-distance distribution function $P(r)$ at 1 mM is much narrower than those computed for the known open-state structures, with a large probability of distribution at ~30 Å (*Figure 2A*). Further, the theoretical scattering profiles for the open-state structure models (*Figure 1—figure supplement 8*) all differ from the experiment curve (*Figure 2—figure supplement 2A*).

## Ensemble refinement of the closed-state structure of $K63-Ub_2$

To characterize the closed-state structure of ligand-free $K63-Ub_2$, we performed rigid-body simulated annealing refinement against the inter-subunit PREs. The linker between the two subunits (Lys63 side chain in the proximal unit and C-terminal flexible residues 72–76 of the distal unit) was given full torsional freedom. A grid search was performed by varying the number of conformers representing

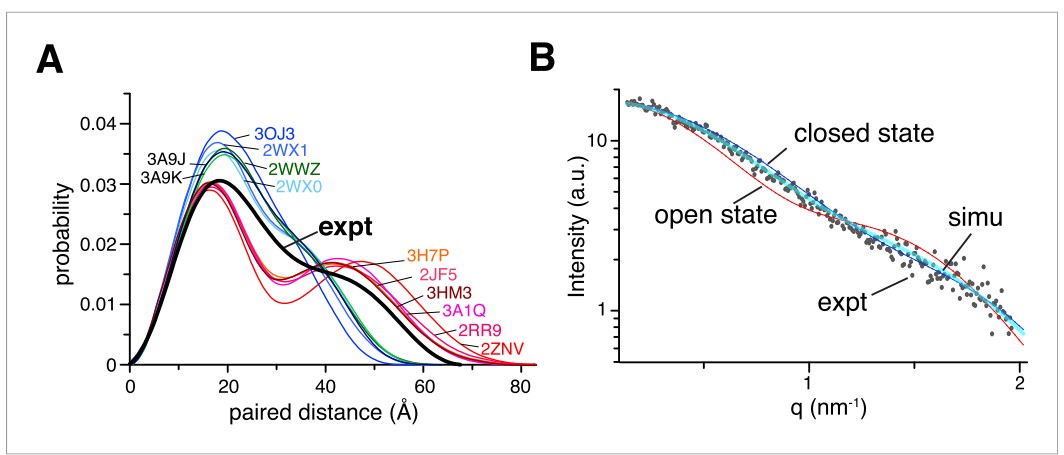

**Figure 2**. Small angle X-ray scattering (SAXS) analysis of ligand-free $K63-Ub_2$. (**A**) Paired-distance distribution curves transformed from the experimental data (black line) or calculated for the known structures of $K63-Ub_2$. Except for the Protein Data Bank (PDB) structures 3H7P, 3HM3, and 2JF5, the bound ligand was removed before calculation. (**B**) Comparison between experimental (gray dots) and simulated scattering data (transparent cyan line), affording a $\chi^2$ value of 1.24. The simulated curve was obtained by linearly combining the theoretical curves calculated for open-state (red line) and closed-state (blue line) solution structures at 30:70 ratio.

The following figure supplements are available for figure 2:

**Figure supplement 1**. Concentration dependence of small angle X-ray scattering (SAXS) profiles for ligand-free $K63-Ub_2$.

**Figure supplement 2**. Comparison between experimental and theoretical scattering curves for (**A**) open-state and (**B**) closed-state structures.

the closed state (from a single conformer to a five-conformer ensemble), and by varying the overall population of the closed state (from 10% to 90%). A single-conformer representation for the closed state does not satisfy the inter-subunit PREs, as assessed by the PRE Q-factor (*Iwahara et al., 2004*). This means that the closed state of ligand-free K63-Ub$_2$ should exist in multiple conformations. The PRE Q-factor rapidly decreases as the number of conformers representing the closed state increases, and levels off with four or more conformers (*Figure 3A*). On the other hand, a closed-state population of at least 30% is required to achieve a good fit to the PRE data (*Figure 3A*). For reasons that will be discussed below, the population of the K63-Ub$_2$ closed state is about 70%. At a 70% population for the closed state with a four-conformer representation, the back-calculated PREs agree well with the experimental ones, affording a PRE Q-factor of 0.22 and correlation coefficient of 0.94 (*Figures 1, 3B*). Importantly, the two paramagnetic conjugation sites, N25C and K48C, provide cross-validating PRE measurements—when refining the ensemble structure of the K63-Ub$_2$ closed state against the N25C data alone, the PRE values predicted for the K48C site largely agree with the experimental values, affording a free Q-factor of 0.46 (*Figure 3—figure supplement 1*). On the other hand, the SAXS profiles computed for the PRE-based closed-state structures differ from the experiment curve, with the calculated intensities larger at scattering angles between 0.5 and 1 nm$^{-1}$ (*Figure 2—figure supplement 2B*).

## Analysis of the closed-state structure of K63-Ub$_2$

To better visualize the ensemble structure of K63-Ub$_2$ in the closed state, we projected the position of the proximal unit relative to the distal unit using spherical coordinates (*Figure 4—figure supplement 1*). Upon reducing the dimensionality, we found that the closed-state structures exist in two clusters, namely C1 and C2 (*Figure 4A*). For each four-conformer structure, one of the conformers falls into C2, while the other three are in C1. The proximal unit of ligand-free K63-Ub$_2$ utilizes distinct interfaces to interact with the distal unit in C1 and C2 states (*Figure 4B,C*), affording buried solvent-accessible surface areas of 283.9 $\pm$ 139.7 Å$^2$ and 200.5 $\pm$ 59.6 Å$^2$, respectively. Significantly, the crystal structures of K63-Ub$_2$ in complex with the ZnF4 domain of A20 (*Bosanac et al., 2010*) and with the NZF domain

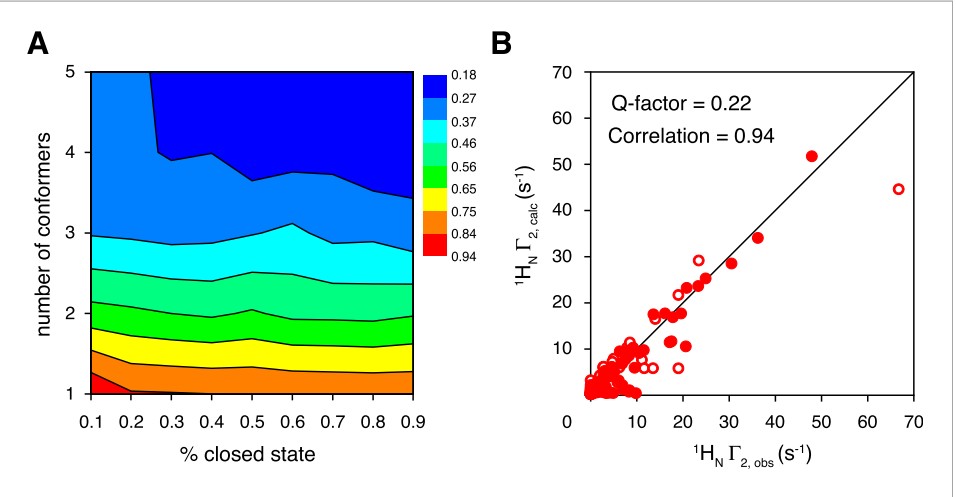

**Figure 3**. Ensemble refinement of the closed-state structure of K63-Ub$_2$ against intra-molecular inter-subunit paramagnetic relaxation enhancements (PREs). (**A**) Heat map of PRE Q-factor upon varying the number of conformers and the population of the closed state. (**B**) The correlation between observed and calculated PREs, with a four-conformer representation at 70% closed-state population. The PRE ensemble Q-factor is 0.26 and 0.18 for the N25C site (open circles) and K48C site (closed circles), respectively, and 0.22 for both sites. The diagonal indicates a perfect correlation to guide the eyes.

The following figure supplement is available for figure 3:

**Figure supplement 1**. Cross-validation of paramagnetic relaxation enhancement (PRE) data.

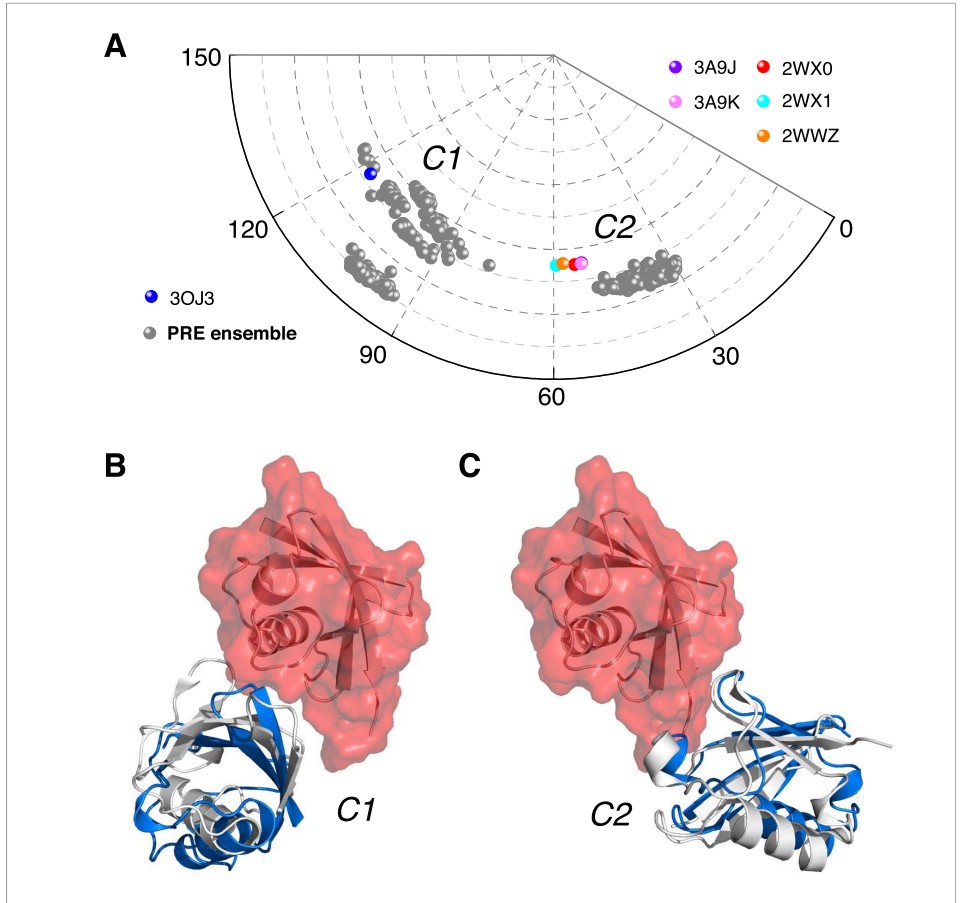

**Figure 4**. Ensemble structure of K63-Ub$_2$ in closed state. (**A**) Projection of the ensemble structures in two dimensions with spherical coordinates. K63-Ub$_2$ crystal structures in closed states are also projected. (**B**, **C**) Comparison of the ligand-free K63-Ub$_2$ structure with K63-Ub$_2$ crystal structures in complex with A20 ZnF4 and NZF TAB2, respectively. With the distal unit superimposed, the other ubiquitin subunit in the crystal structure is shown as a gray cartoon, affording root-mean-square (RMS) differences of 3.93 Å and 1.68 Å for C1 and C2 closed states, respectively.

The following figure supplements are available for figure 4:

**Figure supplement 1**. Definition of the spherical coordinate system.

**Figure supplement 2**. Inter-subunit paramagnetic relaxation enhancements (PREs) predicted from the crystal structures of K63-Ub$_2$ in the closed state.

of TAB2 or TAB3 (*Kulathu et al., 2009*; *Sato et al., 2009b*) are found within or near the C1 and C2 clusters, respectively (*Figure 4A*). The root-mean-square difference (RMSD) between the conformers in C1 and the ZnF4-bound structure of K63-Ub$_2$ is as small as 3.93 Å (*Figure 4B*), while the RMSD between C2 conformers and the NZF-bound structure is as small as 1.68 Å (*Figure 4C*). We predicted the inter-subunit PREs for two known complex structures in closed states (*Figure 4—figure supplement 2A,B*). Linearly combining the two sets of PREs at a 3:1 ratio and 70% total population, the resulting PREs agree well with the experimental values, although some details differ (*Figure 4—figure supplement 2C*). On the other hand, the paired-distance distribution profiles computed for A20 ZnF4 and TAB2/TAB3 NZF complexed K63-Ub$_2$ (with bound ligand removed) display narrower distributions compared to those computed for the open-state structures or to the experimental data (*Figure 2A*).

Taken together, the ensemble structure refinement revealed that in addition to the open state, ligand-free K63-Ub$_2$ also exists in at least two distinct closed states at a significant combined

population. As C1 is represented with multiple conformers, it is possible to further partition the closed state into more conformational states. As the ligand-bound closed structures are similar to ligand-free structures of K63-Ub$_2$ in either C1 or C2 state, a cognate ligand of K63-Ub$_2$ can be preferentially recognized and accommodated into one of the preexisting conformations. As there are some differences between the ligand-free and ligand-bound K63-Ub$_2$ in the closed state (**Figure 4B,C**), the binding may require some induced fit, especially towards the end of the binding process. On the other hand, the open-state conformation of K63-Ub$_2$ may specifically recognize its corresponding ligand like Rap80 tUIM (**Sekiyama et al., 2012**) via a conformational selection mechanism.

## Perturbation of K63-Ub$_2$ conformational space

How does K63-Ub$_2$ inter-convert among the preexisting conformations? To address this, we introduced a charge reversal mutation to residue Glu64 in the proximal unit, resulting an E64R$_P$ mutant of K63-Ub$_2$. Glu64 is located at the interface between the two subunits in both C1 and C2 closed states, opposing the positively charged residues Arg72 and Arg74 in the distal unit (**Figure 5—figure supplement 1**). We reasoned that this mutation should affect the conformational space of K63-Ub$_2$. Indeed, the E64R$_P$ mutation results in chemical shift perturbations (CSPs) in the K63-Ub$_2$ distal unit (**Figure 5—figure supplement 2**). Although the perturbations are small, almost the same residues are perturbed upon E64R$_P$ mutation as upon the covalent linkage of ubiquitin monomers (**Figure 5A** and **Figure 1—figure supplement 1D**). However, the NMR peaks for the perturbed residues in the mutant do not simply move in the direction towards the chemical shift values of the ubiquitin monomer. This can be either due to altered non-covalent interactions around the mutation site, or to a change in the relative population of the conformational states. Therefore, it is difficult to quantitate the CSPs in terms of K63-Ub$_2$ structural change.

Thus we measured the intra-molecular inter-subunit PREs for the E64R$_P$ mutant of K63-Ub$_2$, using the same paramagnetic conjugation scheme. The PRE profile of the mutant is similar to that of the wild type, indicating similar ensemble structures for the mutant (**Figure 5B**). However, the magnitude of the PRE decreases by about 50%. As the inter-subunits' PRE arises only from the closed-state structures of K63-Ub$_2$, smaller PREs indicate that the E64R$_P$ mutation destabilizes the closed state, reducing the closed-state population to half of that of the wild type. At the same time, the E64R$_P$ mutation should result in an increase in the population for the open state.

## Interactions between K63-Ub$_2$ and its ligands

In a conformational selection mechanism, a K63-Ub$_2$ ligand (tUIM, NZF, or ZnF4) is preferentially recognized by one of the preexisting conformational states. Since the relative populations of the conformational states are perturbed upon the E64R$_P$ mutation, we expect that the binding affinities of K63-Ub$_2$ towards the respective ligands differ. Importantly, the mutation is away from the binding interfaces between K63-Ub$_2$ and its ligands (**Figures 6A,B, 7**), and therefore should not directly affect the interactions between K63-Ub$_2$ and its ligands.

Using isothermal calorimetry (ITC), we evaluated the binding affinities between wild type K63-Ub$_2$ and tUIM, and between wild type K63-Ub$_2$ and NZF. The respective $K_D$ values are 9.7 $\pm$ 0.3 μM and 12.2 $\pm$ 0.6 μM (**Figure 1—figure supplement 3**), which agree with the literature values (**Kulathu et al., 2009**; **Sekiyama et al., 2012**). We attempted to measure the binding affinity between A20 ZnF4 domain and K63-Ub$_2$. However, the heat was too small to be fitted (**Figure 7—figure supplement 1**). Thus we resorted to NMR titration for the $K_D$ measurement. Upon titrating A20 ZnF4, a large set of residues in the distal unit of K63-Ub$_2$ was perturbed (**Figure 7A**). Among the perturbed residues, residues 50–62 are located at the interface with ZnF4 (**Bosanac et al., 2010**), and their peaks disappear upon A20 ZnF4 titration, indicating a slow exchange between ZnF4-free and bound species (**Figure 7—figure supplement 2**). We were only able to fit the CSPs at an elevated temperature (313 K instead of 303 K), and determined the $K_D$ value at 384.1 $\pm$ 39.4 μM. A number of other residues in the K63-Ub$_2$ distal unit also experience CSPs upon A20 ZnF4 titration. However, their peaks shift progressively at increasing ZnF4 concentrations, which indicates a fast exchange and should belong to a separate binding event.

Upon the E64R$_P$ mutation, the binding between K63-Ub$_2$ and the open-state ligand tUIM becomes tighter, with the $K_D$ value decreasing by more than fourfold to 2.2 $\pm$ 0.1 μM (**Figure 6C**). On the other hand, the binding towards closed-state ligands weakens—the $K_D$ value of K63-Ub$_2$ binding towards

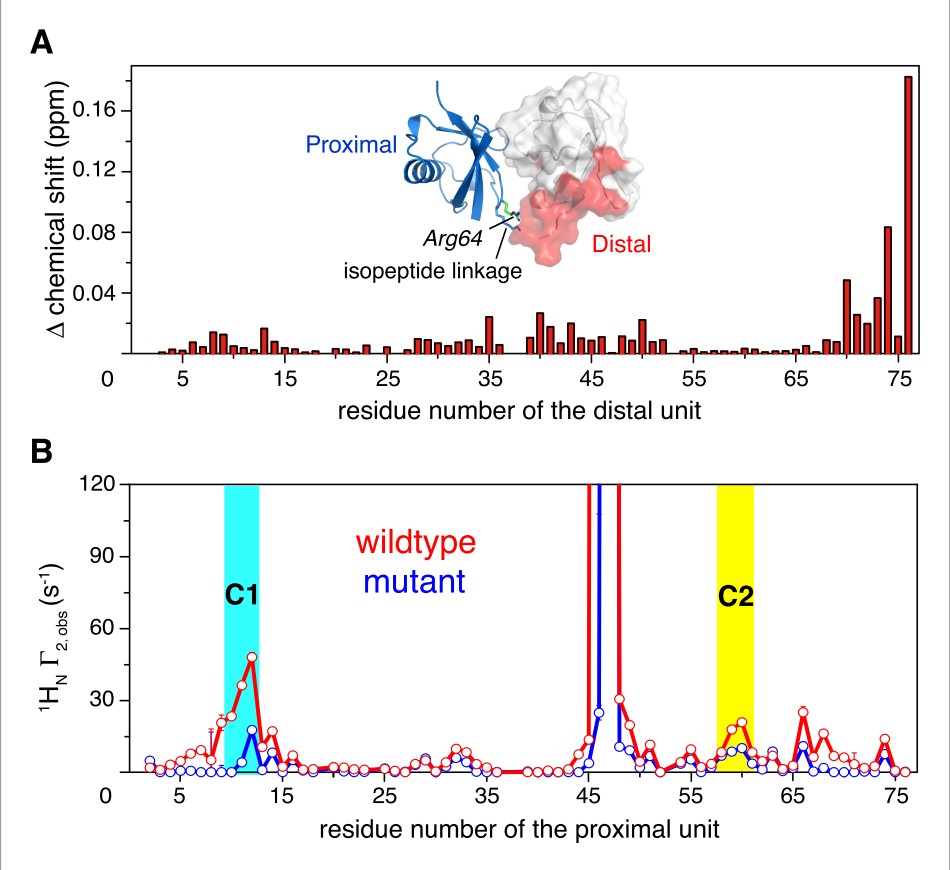

**Figure 5**. Changes in NMR parameters for K63-Ub$_2$ upon E64R$_P$ mutation. (**A**) Chemical shift differences of the distal unit upon mutation. Inset, residues with relatively large chemical shift differences (>0.01 ppm) are mapped to the surface of the distal unit (colored red). (**B**) Decreases in intra-molecular inter-subunit PREs upon mutation with an MTS paramagnetic probe conjugated at K48C site. Lines simply connect the data points. PRE values characteristic of C1 and C2 states are indicated with cyan and yellow strips, respectively. Error bars indicate the SD in PRE measurements.

The following figure supplements are available for figure 5:

**Figure supplement 1**. Structural basis for the perturbation of K63-Ub$_2$ conformational space upon E64R$_P$ mutation.

**Figure supplement 2**. Overlay of 2D NMR spectra for wild type K63-Ub$_2$ and E64R$_P$ mutant at 50 μM with distal unit $^{15}$N-labeled.

NZF increases by ∼50% to 17.8 ± 1.1 μM (**Figure 6D**). Importantly, the enthalpy change ΔH values are almost identical for the bindings involving the wild type and mutant proteins (**Figure 6** and **Figure 1—figure supplement 3**). For the interaction between K63-Ub$_2$ mutant and A20 ZnF4, more peaks at the interface (residues 50–62 of the distal unit) can be traced, which can be attributed to a faster exchange than that of the wild type. Significantly, the $K_D$ value increases by almost threefold to 1199.9 ± 104.9 μM (**Figure 7**). Together, the binding affinity towards the open-state ligand increases at the expense of the binding affinities towards the closed-state ligands, and the changes in binding affinities are caused entropically due to the perturbation of K63-Ub$_2$ conformational space.

The difference in binding affinity can be accounted for by the difference in the conformational energy of K63-Ub$_2$. Based on the PRE measurement, the closed-state population decreases by ∼50% upon E64R$_P$ mutation (**Figure 5**). At the same time, the population for the open state increases by the same amount. Thus, the gain in conformational energy will be manifested as the free energy difference for the increase in binding affinity between K63-Ub$_2$ and its open-state partner tUIM

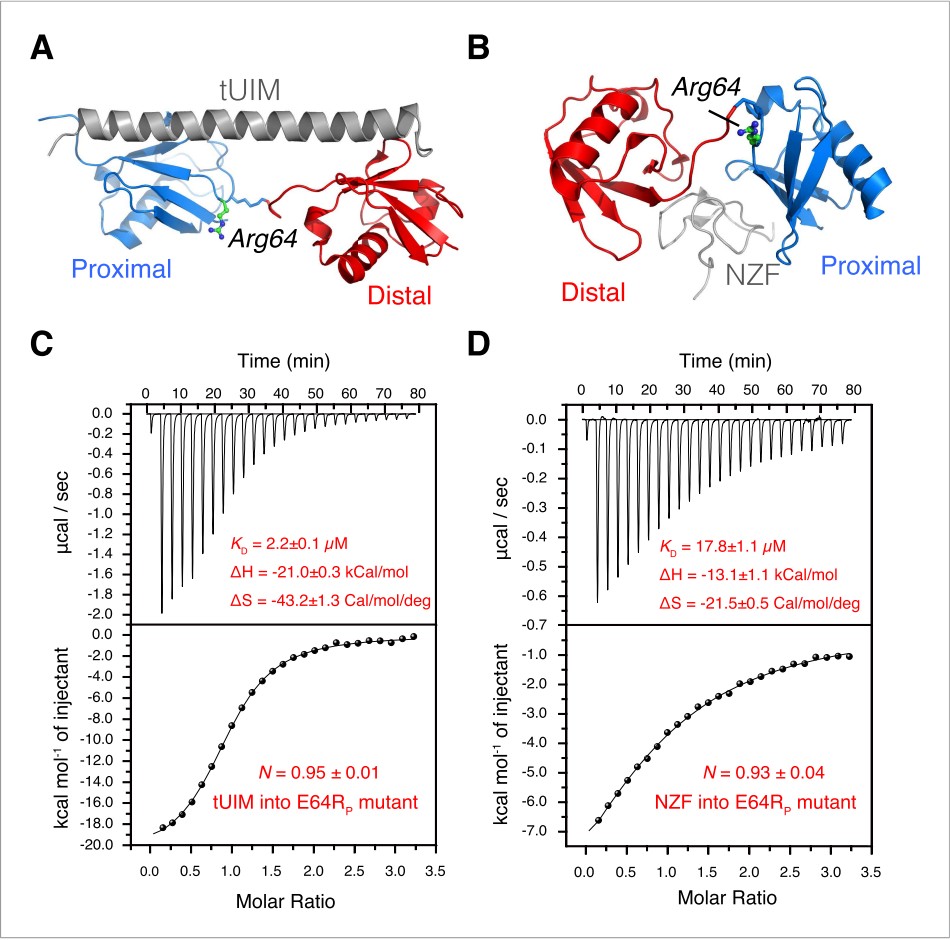

**Figure 6**. The interactions between K63-Ub$_2$ mutant with tUIM and with NZF. (**A**, **B**) Illustration of the E64R$_P$ mutation in the complex structures (Protein Data Bank [PDB] codes 2RR9 and 2WX0). The point mutation is distant from the K63-Ub$_2$ interfaces with tUIM and NZF. (**C**, **D**) Isothermal calorimetry (ITC) measurements for the bindings with tUIM and NZF. The raw data (top panels) are converted to heat per injection, and the fitted curves using one-site binding model are shown as solid lines (bottom panels). The binding affinities $K_D$, binding enthalpy changes $\Delta H$, and entropy changes $\Delta S$ values are averaged over four independent experiments.

($-0.89 \pm 0.04$ kCal/mol). On the other hand, the C1 and C2 closed states forfeit $0.71 \pm 0.08$ kCal/mol and $0.23 \pm 0.09$ kCal/mol in binding free energies, respectively. Our calculation indicated that only when the closed-state population decreases from ~70% to ~35% upon the mutation, in which C1 state population decreases from ~52.5% to ~22.5% and C2 state population decreases from ~17.5% to ~12.5%, could the binding free energy change be the same as the conformational energy change. Indeed, upon the point mutation, a larger decrease was observed for the PRE corresponding to the C1 state than the PRE for the C2 state (**Figure 5B**), and the binding affinity of K63-Ub$_2$ towards a C1 state ligand decreases more than the affinity towards a C2 state ligand (**Figures 6, 7** and **Figure 1—figure supplement 3**). Although the SAXS data have insufficient resolution to resolve multiple closed states, and may include contributions from high-order non-covalent oligomers, linearly combining the SAXS data calculated for the closed and open states at 70% and 30% and without further refinement, we were able to recapitulate the experimental SAXS data with a $\chi^2$ value of 1.24 (**Figure 2B**).

## Discussion

In this study, we have shown that about 70% of K63-Ub$_2$ exists in the closed state, whereas only about 30% of the protein exists in the open state. Our findings disagree with many previous structural

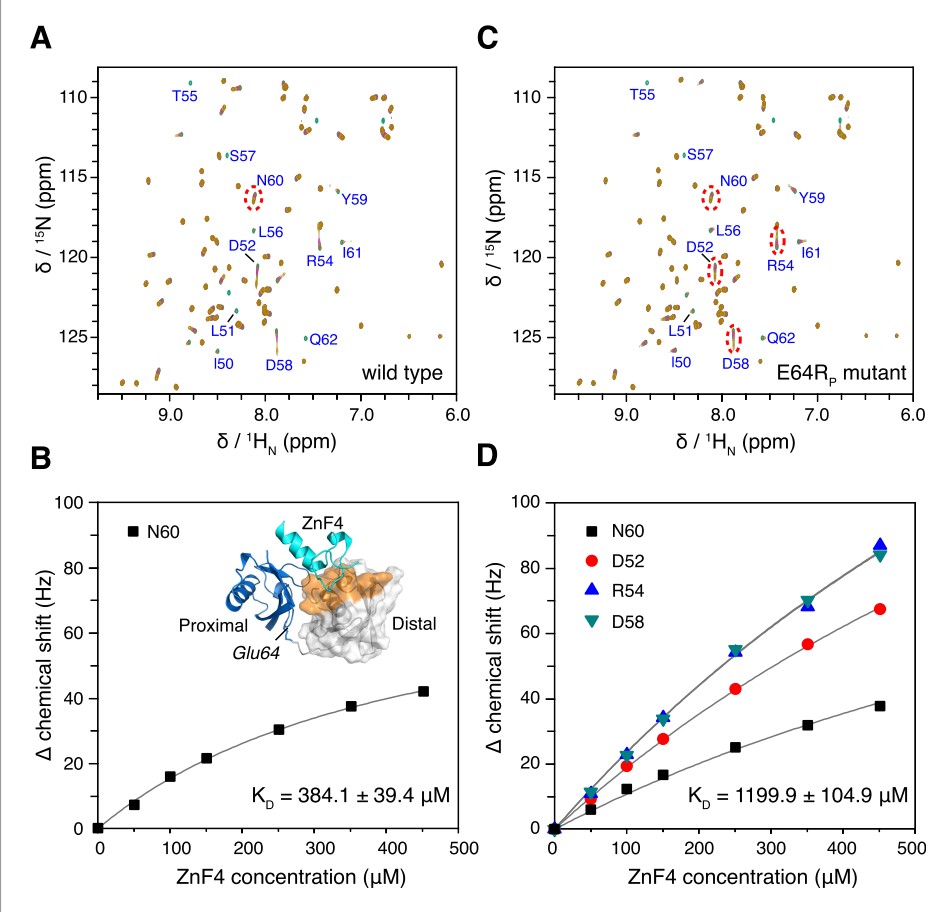

**Figure 7**. The interaction between K63-Ub₂ and A20 ZnF4 at 313 K. (**A**, **C**) NMR titrations of ZnF4 into wild type and mutant K63-Ub₂ with ¹⁵N-labeling at the distal unit. Residues 50–62 (labeled) experience slow timescale exchange and gradually disappear upon ZnF4 titration. (**B**, **D**) Fittings of chemical shift perturbations to binding isotherms. The chemical shift perturbations are calculated as $(\Delta\delta_H{}^2 + \Delta\delta_N{}^2)^{0.5}$ in Hz units. Inset, ZnF4-binding surface on K63-Ub₂ distal unit (residues 50–62) is mapped (colored orange).

The following figure supplements are available for figure 7:

**Figure supplement 1**. Isothermal calorimetry (ITC) measurements for the bindings between A20 ZnF4 and (**A**) wild type and (**B**) E64R_P mutant of K63-Ub₂ proteins.

**Figure supplement 2**. Overlay of NMR spectra for K63-Ub2 and K63-Ub2 mixed with equimolar A20 ZnF4 at (**A**) 303 K and (**B**) 313 K.

characterizations of K63-Ub₂ (*Varadan et al., 2004*; *Datta et al., 2009*; *Dikic et al., 2009*; *Komander et al., 2009*; *Weeks et al., 2009*; *Komander and Rape, 2012*), which reported that ligand-free K63-Ub₂ exists only in the open state. Nevertheless, the closed-state population is in line with our previous finding that ubiquitin monomer non-covalently dimerizes with an apparent $K_D$ value of $4.9 \pm 0.3$ mM (*Liu et al., 2012*). In the non-covalent dimer of ubiquitin monomer, a ubiquitin adopts an array of orientations in respect to the other. With a covalent linkage, the non-covalent interaction between ubiquitin becomes intra-molecular and restricted. Therefore, a significant population of diubiquitin should exist in the closed compact conformation regardless of the ubiquitin linkage. Indeed, studies have indicated that K48-Ub₂ mainly exists in the closed state (*Cook et al., 1992*, *1994*; *Phillips et al., 2001*; *Varadan et al., 2002*; *Eddins et al., 2007*; *Hirano et al., 2011*; *Ye et al., 2012*). For ligand-free K63-Ub₂, however, only a recent single-molecule fluorescence resonance energy transfer (smFRET) study has provided direct experimental evidence and indicated that the

protein exists in a closed state at a population of ~75% (*Ye et al., 2012*). Here, using paramagnetic NMR, SAXS, and mutational analysis, we found that the population of the closed state of ligand-free K63-Ub$_2$ is ~70%. But why have only open-state structures been reported for ligand-free K63-Ub$_2$? A possible explanation is that the open-state structure is more readily captured owing to non-covalent interactions between neighboring unit cells, and becomes enriched during crystallization processes.

Further, we have identified two distinct closed states, namely C1 and C2, with different populations (*Figure 4A*). The PRE NMR provides $1/r^6$ ensemble-averaged distance information (*Clore and Iwahara, 2009*). So the inverse problem is to determine the constituting conformational states that give rise to the ensemble-averaged PRE observables. Here by projecting the structures with spherical coordinates, we were able to visualize the distinct conformational states of ligand-free K63-Ub$_2$. In comparison, the smFRET study on ligand-free K63-Ub$_2$ (*Ye et al., 2012*) measured just a single distance between the N-terminus of the distal unit and the C-terminus of the proximal unit, and could not distinguish multiple closed states or provide a structural description for each state. Similarly, SAXS analysis is unable to reveal how two nearly globular proteins are docked to each other in atomic detail (*Figure 2A*).

Our structural analysis based on the PRE revealed that the C1 and C2 closed states utilize different binding interfaces. Importantly, the ligand-free structures of K63-Ub$_2$ are similar to the respective ligand-bound structures (*Kulathu et al., 2009*; *Sato et al., 2009a*; *Bosanac et al., 2010*) (*Figure 4B,C*). This suggests a conformational selection mechanism for K63-Ub$_2$ target recognition. This mechanism is further supported by conformational energy analysis. The inter-conversion between K63-Ub$_2$ conformational states and ligand binding are coupled equilibria, and the population for each conformational state weights on the binding affinity towards a respective ligand. For the interaction between K63-Ub$_2$ and tUIM, an open-state ligand, the change in binding free energy accompanying an E64R$_P$ mutation can be fully accounted for by the increase in the relative population of the open state. On the other hand, the change in conformational energy also accounts for the difference in binding affinities between wild type and mutant K63-Ub$_2$ towards closed-state ligands TAB2/TAB3 NZF and A20 ZnF4.

Together, our ensemble structural refinement and mutational analysis revealed that ligand-free K63-Ub$_2$ adopts at least three conformational states, including one open state and two closed states, each of which can accommodate cognate ligands. Closed compact structures have been reported for ligand-free diubiquitins with Lys48 (*Cook et al., 1992*), Lys11 (*Matsumoto et al., 2010*; *Castaneda et al., 2013*), Lys29, and Lys33 (*Kristariyanto et al., 2015*; *Michel et al., 2015*) linkages. These structures are different from the C1 or C2 closed-state conformations of K63-Ub$_2$, and therefore are involved in different functions. As such, a covalent ubiquitin linkage dictates how the two subunits non-covalently interact with each other in a diubiquitin, and resulting quaternary arrangements encode specific cell signals. For K63-Ub$_2$, the open state recognizes tUIM of Rap80 and is involved in DNA damage repair, the C2 closed state recognizes the NZF domain of TAK1 binding proteins and is involved in the activation of NF-κB signaling, while the C1 state recognizes the ZnF4 domain of A20 and is involved in the termination of NF-κB signaling pathways (*Figure 8*). Constructed from repeating units of diubiquitins, a polyubiquitin should exist in a combination of quaternary structures of the diubiquitins and participate in diverse functions.

## Materials and methods

### Sample preparation

Human ubiquitin was cloned into a pET11a vector; single-point mutations including N25C, K48C, E64R, K63R, and 77D were introduced using QuikChange (Stratagene). BL21 star cells were used for protein expression, and were grown in either LB medium (for preparing unlabeled proteins) or in M9-minimum medium (for preparing isotope-enriched proteins). All ubiquitin proteins were purified on Sepharose SP and Sephacryl S100 columns (GE Healthcare, Piscataway, NJ) in tandem. Ligation between two ubiquitin molecules was based on the established protocol (*Pickart and Raasi, 2005*). Briefly, the proximal unit carrying a 77D mutation (Asp77 appended at the C-terminus) was mixed equimolarly with the distal unit carrying a K63R mutation, to thus ensure a single ligation product. With the addition of 2.5 μM E1 and 10 μM E2 (Mms2/Ubc13 complex from yeast), 2 mM ATP, and 5 mM MgCl$_2$, ligation between two ubiquitin monomers was allowed to proceed for 5 hr at 30°C. The reaction was quenched with 5 mM DTT and 2 mM EDTA. The product was purified on a Sephacryl S100 column.

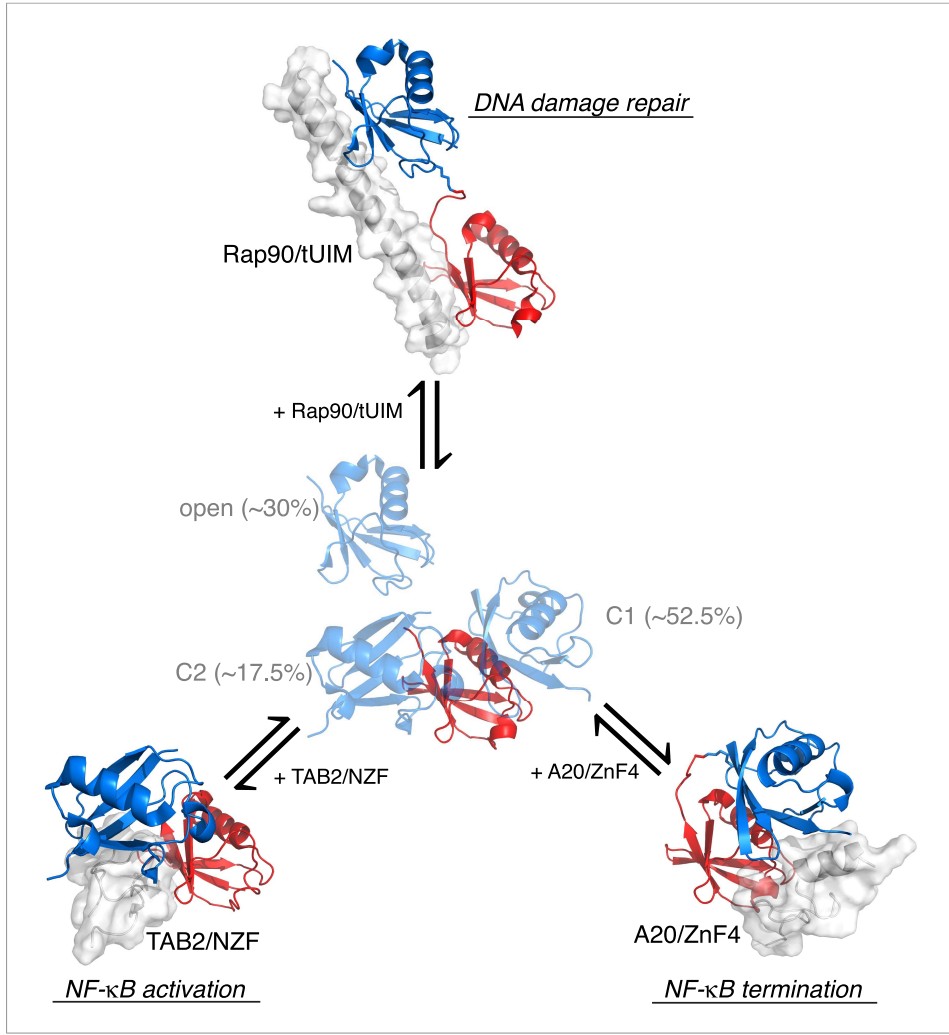

**Figure 8**. Proposed mechanism for K63-Ub$_2$ signaling. In the absence of a ligand, K63-Ub$_2$ alternates between an open and two closed states. A specific ligand can be accommodated and bound to one of the three preexisting conformations, eliciting the downstream signal.

Tandem ubiquitin-interacting motif (tUIM) from human protein Rap80 encompassing residues 79–124, the fourth ZnF4 domain of A20 (residues 590–635), and NZF domain from human TAK1-binding protein 2 (TAB2, residues 663–693) were cloned into the pGEX vector. Since tUIM has no 280 nm UV absorption, a C121Y mutation was introduced. It has been shown that modification to this residue has no effect on the structure of tUIM or the interaction between tUIM and K63-Ub$_2$ (*Sekiyama et al., 2012*). All three proteins were expressed in BL21 star cells in LB medium. The proteins were purified off a GST affinity column. With the GST tag removed by TEV protease, the proteins were further purified through a Sephacryl S100 column. For the purification of ZNF4 and NZF, the buffer also contains 5 mM DTT and 50 µM ZnCl$_2$. All purified proteins were confirmed by electrospray mass spectrometry (Bruker Daltonics, Germany).

## NMR data collection

A single-point cysteine mutant of K63-Ub$_2$, either N25C or K48C in the distal unit, was reacted with a fivefold excess of S-(2,2,5,5-tetramethyl-2,5-dihydro-1H-pyrrol-3-yl) methyl methanesulfonothioate (MTS; from Toronto Research Chemicals, Canada) or a fourfold excess of [N-(2-maleimido)ethyl] ethylenediamine-N,N,N′,N′-tetraacetate (pre-incubated with a twofold excess of MnCl$_2$, EDTA-Mn$^{2+}$) for 3 hr at room temperature. Unreacted probe was removed by desalting. The conjugation product was

confirmed by mass spectrometry for a mass difference of 184 Da for MTS and 414 Da for EDTA conjugation. The NMR buffer contains 100 mM NaCl, 10 mM sodium acetate at pH 6.0, and 10% $D_2O$. The paramagnetic NMR data were collected on a 500 µM sample at 303 K on Bruker 850 MHz or 600 MHz instruments, each equipped with a cryogenic probe. Transverse relaxation rates of amide protons for the $^{15}N$-labeled subunit of K63-Ub$_2$ protein were measured using the standard pulse scheme with a 4 ms delay between the two time points, $T_a$ and $T_b$ (*Iwahara et al., 2007*). Inter-molecular PREs were measured for a 500 µM equimolar mixture of K63-Ub$_2$, with paramagnetic conjugation and isotope labeling on two separate proteins. The peak intensity at the second time point $T_b$ is given as follows:

$$I_b = I_a \exp[-R(T_b - T_a)], \tag{1}$$

in which the relaxation rates $R$ can be diamagnetic relaxation rates $R_2$, or $(R_2 + \Gamma_{2,inter})$ for the equimolar mixture, or $(R_2 + \Gamma_{2,inter} + \Gamma_2)$ for the paramagnetic sample. $I_a$ and $I_b$ are the peak intensities at $T_a$ and $T_b$. Using the equimolarly mixed sample as the PRE reference, the intra-molecular inter-subunit PRE $^1H$ $\Gamma_2$ value can be determined as follows:

$$\Gamma_2 = \frac{1}{T_b - T_a} \ln \frac{I_{inter}(T_b)I_{para}(T_a)}{I_{inter}(T_a)I_{para}(T_b)}. \tag{2}$$

The same scheme was used for determining intra-molecular inter-subunit PREs for the E64R$_P$ mutant of K63-Ub$_2$. At 50 µM concentration, the percentage of K63-Ub$_2$ dimer can be negligible and the $\Gamma_{2,inter}$ term disappears. Thus the peak intensities for a single time point measurement scheme are defined as:

$$I_{dia,50} = I_0 \exp[-R_2 T], \tag{3}$$

$$I_{para,50} = I_0 \exp[-(R_2 + \Gamma_2)T], \tag{4}$$

in which $I_0$ is the intensity at the beginning of the pulse sequence, and $T$ is ~9.2 ms (*Iwahara et al., 2007*). Owing to the $\Gamma_{2,inter}$ term, peak intensities at 500 µM concentration are calculated as below:

$$I_{para,500} = I_0 \exp\left[-(R_2 + \Gamma_{2,inter} + \Gamma_2)T\right], \tag{5}$$

$$I_{inter,500} = I_0 \exp\left[-(R_2 + \Gamma_{2,inter})T\right], \tag{6}$$

in which $I_{para,500}$ and $I_{inter,500}$ are peak intensities for 500 µM paramagnetic sample and 500 µM equimolar mixture, respectively. Taking the ratios of the four equations above, the following relationship can be obtained:

$$\frac{I_{dia,50}}{I_{para,50}} = \exp(\Gamma_2 T) = \frac{I_{inter,500}}{I_{para,500}}. \tag{7}$$

## Ensemble structure refinement and analysis

Refinement against experimental restraints was conducted using Xplor-NIH (*Schwieters et al., 2006*). A three-conformer representation for each paramagnetic probe at each conjugation site (N25C or K48C) was employed, to thus recapitulate the conformational flexibility for the paramagnetic probe (*Iwahara et al., 2004*). With the protein backbone fixed, the dihedral angles for the rotatable bonds between the paramagnetic center and the protein backbone were optimized. Excluding structures with large van der Waals violations, intra-molecular and intra-subunit PREs were calculated for each structure, which were employed as the target values to restrain the spatial distribution of the paramagnetic probes in the subsequent calculations.

The starting coordinates of each ubiquitin subunit were taken from Protein Data Bank (PDB) structure 1UBQ (*Vijay-Kumar et al., 1987*), and two ubiquitin molecules were patched together with an isopeptide bond between the Lys63 side chain of the proximal unit and the C-terminus of the distal unit. While keeping the coordinates of the distal unit fixed (conjugated with the paramagnetic probes at N25C and K48C sites, each in three-conformer representation), the proximal unit of K63-Ub$_2$ was treated as a rigid body that can rotate and translate as a whole. The Lys63 side chain of the proximal unit and residues 72–76 of the distal unit were given full torsional freedom.

To initiate the ensemble rigid-body simulated annealing, the coordinates for the proximal unit (also including residues 72–76 of the distal unit) were replicated to make additional members of the

ensemble. Each ensemble member was subjected to random rotational and translational movement, and was allowed to overlap. The simulated annealing ensemble refinement was performed with a target function that comprises the inter-subunit PRE restraints for residues 1–71, theoretical intra-subunit PRE restraints to confine the spatial distribution of the paramagnetic probes, the van der Waals repulsive term, and covalent energy terms. Square-well energy potential was used for the PRE term—no energy penalty was given when the back-calculated PRE value was within $\pm$ the experimental error of the target value. Residues that are completely broadened out in the paramagnetic spectrum were given a large PRE target value with the lower bound at 120 s$^{-1}$. An apparent PRE correlation time ($\tau_c = 7.2$ ns) was estimated based on the rotational correlation time of the diamagnetic protein ($\sim$7.6 ns for the closed state) and the large electron relaxation time of the nitroxide spin radical ($\sim$150 ns) (*Tang et al., 2007*). The population of the closed state was varied from 10% to 90% in 10% increments, which was implemented as a scaling factor for the back-calculated PRE value. In simulated annealing refinement, the PRE energy force constant was ramped from 0.01 to 1 kcal mol$^{-1}$ s$^2$, and the temperature was cooled from 3000 to 25 K. For each combination of closed-state population and number of conformers, 160 structures were calculated. The agreement between the observed and calculated PRE rates was assessed with PRE Q-factor for both conjugation sites (*Iwahara et al., 2004*). To better visualize the structures, a spherical coordinate system was constructed, with the origin set at the center-of-mass of the distal unit. Analysis of the buried interfaces and rendering of an atomic probability map (*Schwieters and Clore, 2002*) were performed using Xplor-NIH (*Schwieters et al., 2006*). Structure figures were illustrated using PyMOL (The PyMOL Molecular Graphics System, Version 1.7; Schrödinger, LLC).

## SAXS measurement

Solution SAXS was performed at 303 K on the SAXSess mc$^2$ platform (Anton Paar, Graz, Austria) equipped with a sealed-tube X-ray source and a CMOS diode array detector. The proteins were extensively dialyzed to the same buffer used for NMR, and the SAXS profile for the matching buffer was recorded for background subtraction. To remove high molecular weight aggregate, the protein samples were centrifuged at 15,000 rpm for 30 min prior to each experiment, and the upper portion of the supernatant (the concentration was measured again at UV 280 nm) was pipetted and loaded into a quartz cuvette. The sample was placed 306 mm from the detector with a slit width of 10 mm. The SAXS data were collected in 30 min increments for a total of 5 hr (10 hr for the 0.5 mM sample); no difference was found between the first and last frames of SAXS data. The maximum distance of the particle ($D_{max}$) was extrapolated from the paired-distance distribution function $P(r)$ after indirect Fourier transformation of the $I(q)$ scattering curve. The data collected at 1 mM were used for further analysis. The theoretical $P(r)$ curve was calculated for each known structure of K63-Ub$_2$ using CPPTRAJ in the AMBER 14 package (UCSF). The bound ligand was removed if present, and any missing residues from the crystal structure were patched using Xplor-NIH (*Schwieters et al., 2006*). With a water layer of $\sim$3.5 Å thickness padded to protein structure (291–349 explicit water molecules added depending on the PDB structure), the calculation of paired-distance distribution was performed at 0.5 Å resolution. The theoretical $P(r)$ curve was smoothed using a 10-point spline function for plotting, and was normalized to a total area of 1. The theoretical scattering $I(q)$ profiles were calculated using CRYSOL (*Svergun et al., 1995*) without fitting to the experimental data, and were scaled by the first point of the experimental scattering data.

## Mutational and binding analyses

The isothermal calorimetry (ITC) binding experiment was performed on a VP-ITC instrument (GE Healthcare) at 303 K. All protein samples were prepared in the same buffer as in the NMR experiments. A 20 μM sample of K63-Ub$_2$ protein, either wild type or E64R$_P$ mutant, was placed in the reservoir. The titrant, 300 μM tUIM, TAB2 NZF, or A20 ZnF4 proteins, was titrated into K63-Ub$_2$ drop-wise. Dilution heat was subtracted by titrating tUIM or NZF into a matching buffer and was measured for each experiment. After converting to heat per injection, the curves were fitted using a one-site binding model using Origin 8.1 software. All ITC titrations were performed at least four times. NMR titration of A20 ZnF4 was performed on the Bruker 850 MHz instrument at 303 K or 313 K, by titrating 50, 100, 150, 250, 350, or 450 μM ZnF4 into wild type or E64R$_P$ mutant K63-Ub$_2$ protein (distal unit $^{15}$N-labeled). The exchange timescale at 313 K is faster than at 303 K, which allowed us to fit the binding isotherm from the chemical shift perturbations of residues at the ZnF4 binding interface.

Based on the binding affinities measured by ITC and by NMR, the binding free energies differ by $-0.89 \pm 0.04$, $0.23 \pm 0.09$, and $0.71 \pm 0.08$ kCal/mol upon $E64R_P$ mutation, for the bindings towards tUIM, NZF, and ZnF4, respectively. The probability of open state $P_o$ can be defined,

$$P_o = \frac{e^{-\varepsilon_o/k_BT}}{1 + e^{-\varepsilon_o/k_BT}},$$ (8)

in which $\varepsilon_o$ is the free energy of the open state relative to the closed state, and $k_B$ is the Boltzmann constant. Thus, the energy for the open state can be calculated:

$$\varepsilon_o = k_BT \ln(1/P_o - 1).$$ (9)

The difference in conformational energy for the open state between the mutant and wild type $K63\text{-}Ub_2$ can be calculated as below:

$$\Delta\Delta G_{conformation} = (\varepsilon_{o,mt} - \varepsilon_{o,wt})N_A = RT \ln(1/P_{o,mt} - 1) - RT \ln(1/P_{o,wt} - 1),$$ (10)

in which $\varepsilon_{o,mt}$ and $\varepsilon_{o,wt}$ are the conformational energies for the open state of the mutant and of the wild type, respectively, and $N_A$ is the Avogadro constant. As the overall population of the closed state decreases by ~50% for the mutant, as estimated from the PRE, the population of the open state follows this relationship:

$$P_{o,mt} = 1 - 0.5 \times (1 - P_{o,wt}).$$ (11)

As the differences in the binding affinities towards the respective ligands of $K63\text{-}Ub_2$ are caused entropically, the difference in conformational free energy should be equal to the difference in binding free energy. Solving *Equations 10* and *11*, one could determine that the open-state population increases from ~30% for the wild type to ~65% for the mutant, which corresponds to a conformational energy change of $-0.88$ kCal/mol. At the same time, the closed-state population drops from ~70% to ~35%. Further, the ratio between C1 and C2 closed states can be determined at about 3:1 for the wild type $K63\text{-}Ub_2$. Thus, upon the point mutation, the population of C1 state drops from 52.5% to 22.5 %, and the population of C2 state drops from 17.5% to 12.5 %, which correspond to conformational energy changes of 0.80 and 0.24 kCal/mol, respectively. Such changes are in line with the design of the $E64R_P$ mutant, and are also consistent with the relative decreases in PRE values.

## Acknowledgements

We thank the Chinese Ministry of Science and Technology (2013CB910200) and the National Natural Science Foundation of China (31225007 and 31170728) for grant support. The research of CT was supported in part by an International Early Career Scientist grant from the Howard Hughes Medical Institute. ZL was supported in part by China Postdoctoral Science Foundation.

## Additional information

### Funding

| Funder | Grant reference | Author |
| --- | --- | --- |
| Ministry of Science and Technology of the People's Republic of China | 2013CB910200 | Chun Tang |
| National Natural Science Foundation of China | 31225007 | Chun Tang |
| National Natural Science Foundation of China | 31170728 | Chun Tang |
| Howard Hughes Medical Institute (HHMI) | International Early Career Scientist | Chun Tang |
| China Postdoctoral Science Foundation | 2015M571860 | Zhu Liu |

The funders had no role in study design, data collection and interpretation, or the decision to submit the work for publication.

## Author contributions
ZL, ZG, Acquisition of data, Analysis and interpretation of data, Drafting or revising the article; W-XJ, JY, W-KZ, Acquisition of data, Drafting or revising the article; D-CG, W-PZ, Analysis and interpretation of data, Drafting or revising the article; M-LL, Conception and design, Drafting or revising the article; CT, Conception and design, Analysis and interpretation of data, Drafting or revising the article

## Author ORCIDs
Chun Tang, http://orcid.org/0000-0001-6477-6500

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
