## [Decision Letter]

Thank you for sending your work entitled “Lys63-linked Ubiquitin Chain Adopts Multiple Conformational States for Specific Target Recognition” for consideration at *eLife*. Your article has been favorably evaluated by John Kuriyan (Senior editor) and three reviewers, one of whom, Volker Dötsch, is a member of our Board of Reviewing Editors.

The Reviewing editor and the other reviewers discussed their comments before we reached this decision, and the Reviewing editor has assembled the following comments to help you prepare a revised submission.

The paper presents a thorough and nice analysis of the major conformation space of the dynamic K63-linked diubiquitin. The results are of considerable interest for this field, and present also a nice example for a broader community.

Technically, the work is mostly sound and well done. However, there are a number of concerns and technical issues that should be addressed before considering publication of the manuscript.

1) The statistical significance and validation of the ensemble selection is not fully clear. The analysis appears biased as simultaneously the number of members in the ensemble and their fractional contribution are varied. The authors simply propose that with four members and a 70% contribution a small Q-factor is observed, but no independent multi-parameter optimization is presented (and may not be possible without additional data). In my view, this is not statistically sound and introduces a bias that represents an overfitting of the data. There is no cross-validation presented (using other experimental data, i.e. with different spin label positions or using other experimental data). It is not clear how the K11Cp or K48Cd PRE data should provide an independent cross-validation, as the data are not shown and interpreted (for cross-validation). Have the authors considered SAXS measurements to provide additional independent data that could help fitting populations if the predicted SAXS data are sufficiently different for the open and closed conformations?

2) Have the authors analyzed chemical shift differences (comparing single and linked ubiquitins) or looked for potential interdomain NOEs that could support the observed interfaces?

3) Complete data for ITC data need to be provided, especially considering the important conclusions the authors derive from them. Fitted stoichiometries and blank titrations (dilution heats) should be shown for all ITC data (including replicates). The ITC data for the UBP NZF (prefers C2 state) are not convincing. The authors state that the mutation decreases the affinity by 50%. However, especially taking into account the fitting error it is much less than 50%. It is also not clear the errors indicated are fitting errors or averages and uncertainties from the replicate measurements, which they should be, given the conclusions based on the rather small difference in binding affinities. The authors mention in the Methods section, that they have measured duplicates. They should provide all the ITC measurements in the supplement so that the quality and validity of their conclusions can be assessed. For NZF, it seems that repeating the measurement three or even four times would be necessary. Also, as the authors say that the mutations should have a stronger impact on the C1 state, it would therefore be informative that the authors do similar ITC measurements and analyses with a C1 ligand, e.g. the mentioned antibody, to further corroborate the conclusions.

4) The authors should illustrate that the spin labels introduced do not directly interfere with either the open or the closed state structures, or potentially increase the tendency towards a closed state structure. This should be shown by highlighting the position of K11/K48 in structural figures (e.g. Figure 3) for K63-Ub_2_ alone, and when to bound to the two ligands investigated.

In addition, the authors should also verify that the spin labels themselves do not affect binding of the ligands, i.e. by comparing ITC data for wild type and spin labeled proteins.

5) The authors propose that the reduced τ_c_ values for the charge reversal mutation indicates a shift to more open conformations. However, it could also be that the charges also reduce the (known) tendency of (di)ubiquitin to dimerize. Can the authors rule out this possibility? Have they measured relaxation data for the monomer harboring the same charge reversals?

6) The Abstract is phrased as addressing a biophysical rather than an important biological/biochemical problem. Also the compact introduction may appeal more to a smaller expert community. On both I would suggest to change the emphasis towards the biochemical result and its possible biological impact, and add a short section to the Introduction and Discussion.

7) What is the structural basis for the effect of the charge reversal mutant?

8) Published literature where multi-domain conformations have been studied should be appropriately referenced, i.e. work by the groups of Berlin/Fushman (theoretical analysis), Blackledge, Luchinat, Sattler, Grzesiek, Kay and others (using PRE, RDC, SAXS data).

[Editors' note: further revisions were requested prior to acceptance, as described below.]

Thank you for resubmitting your work entitled “Lys63-linked Ubiquitin Chain Adopts Multiple Conformational States for Specific Target Recognition” for further consideration at *eLife*. Your revised article has been favorably evaluated by John Kuriyan (Senior editor), a Reviewing editor, and one reviewer. The manuscript has been improved but there are some remaining issues that need to be addressed before acceptance, as outlined below:

1) The authors suggest that the use of SAXS data, which could help fitting populations, is not feasible as K63-Ub_2_, or mono-Ub dimerize non-covalently and this strongly affects the scattering. This would however also jeopardize the overall conclusions as one wonders whether the PRE data might also reflect intermolecular PREs? It would be interesting to show the SAXS data attached to a primary figure and their concentration dependence to judge how strong these effects really are.

2) Concerning the ITC data: The authors should additionally report the stoichiometries fitted for the ITC data. For better comparison *K*_D_ not *K*_A_ values should be reported consistently.

3) The Introduction addresses a general audience now, the Abstract not really yet. It could still be improved:

In short (I have no objection what parts would be used..): The authors show that free K63-Ub_2_ exists as a dynamic ensemble consisting of multiple closed and extended quaternary arrangements. This quaternary structural plasticity of K63-Ub_2_ enables it to be recognised in a variety of signalling pathways. Upon binding to a target protein one of the quaternary arrangements is selected. The authors show that different target proteins or point mutations can select or shift the equilibrium between different quaternary arrangements.

4) Possible error: In the subsection headed “NMR data collection”: [Disp-formula equ3] and [Disp-formula equ4] seem to be partially swapped (Is Γ2 in the dia term correct? If not, [Disp-formula equ7] might also be switched.)

---

## [Author Response]

1) The statistical significance and validation of the ensemble selection is not fully clear. The analysis appears biased as simultaneously the number of members in the ensemble and their fractional contribution are varied. The authors simply propose that with four members and a 70% contribution a small Q-factor is observed, but no independent multi-parameter optimization is presented (and may not be possible without additional data). In my view, this is not statistically sound and introduces a bias that represents an overfitting of the data.

We performed the grid search in the ensemble refinement against inter-subunit PRE data, by varying the number of conformers and the population of the closed state. The plot is presented in Figure 2 of the revised manuscript, which shows that the PRE Q-factor reaches a minimum with at least four conformers at population of >30%. A more accurate percentage for the closed state can be determined based on mutation and energy analysis. The closed-state population of ligand-free K63-Ub_2_ is consistent with both the smFRET study (Ye et al., Nature 2012) and the dimerization tendency of ubiquitin monomer (Liu et al.*,* Angew. Chem. Int. Ed. 2012). Further, we deduced that there are at least two distinct closed states of ligand-free K63-Ub_2_ at about 3:1 molar ratio, and the ligand-free structures are similar to the known ligand-bound structures. We linearly combined the back-calculated PREs for the two ligand-bound structures of K63-Ub_2_ (PDB codes 3OJ3 and 2WX0) with scaling factors of 52.5% and 17.5% (molar ratio of 3:1 with a total population of 70%), respectively. The generated PREs are similar to the experimental data, thus corroborating the robustness of our ensemble refinement against the PRE data. The comparison is presented in Figure 3–figure supplement 2 of the revised manuscript.

There is no cross-validation presented (using other experimental data, i.e. with different spin label positions or using other experimental data). It is not clear how the K11Cp or K48Cd PRE data should provide an independent cross-validation, as the data are not shown and interpreted (for cross-validation).

We performed ensemble structure refinement against only one set of the PRE data (K25C). We were able to cross-validate the PREs at the other conjugation site, K48C. The back-calculated PRE data agree quite well with the experimental data (free Q-factor 0.46). The result is presented in Figure 2—figure supplement 1 of the revised manuscript.

Have the authors considered SAXS measurements to provide additional independent data that could help fitting populations if the predicted SAXS data are sufficiently different for the open and closed conformations?

We did perform SAXS measurement on K63-Ub_2_. It should be noted that K63-Ub_2_ can noncovalently dimerize. The resulting K63-Ub_2_ dimer (or oligomer) has larger molecular weight and scatters more X-ray, even though its population can be quite low. Therefore, interpreting the populations of the constituting conformational states for SAXS would not be straightforward. For the same reason, we used 1:1 equimolar mixture for the PRE reference.

2) Have the authors analyzed chemical shift differences (comparing single and linked ubiquitins) or looked for potential interdomain NOEs that could support the observed interfaces?

The chemical shift differences between individual subunits of K63-Ub_2_ and ubiquitin monomer is very small, as illustrated in Figure 1—figure supplement 1 of the revised manuscript. The chemical shift differences can be either resulted from covalent linkage or due to noncovalent interactions between the two subunits. Therefore we were not able to dissect the contribution of the latter and unambiguously map the interface.

We also attempted to measure inter-subunit NOEs between ^13^C-labeled proximal unit and unlabeled distal unit using a half-filtered scheme, but to no avail. Our data are consistent with the previous NMR studies (45; 44). In the latter work the authors failed to detect cross-saturations between the two subunits.

3) Complete data for ITC data need to be provided, especially considering the important conclusions the authors derive from them. Fitted stoichiometries and blank titrations (dilution heats) should be shown for all ITC data (including replicates). The ITC data for the UBP NZF (prefers C2 state) are not convincing. The authors state that the mutation decreases the affinity by 50%. However, especially taking into account the fitting error it is much less than 50%. It is also not clear the errors indicated are fitting errors or averages and uncertainties from the replicate measurements, which they should be, given the conclusions based on the rather small difference in binding affinities. The authors mention in the Methods section, that they have measured duplicates. They should provide all the ITC measurements in the supplement so that the quality and validity of their conclusions can be assessed. For NZF, it seems that repeating the measurement three or even four times would be necessary.

For each ITC titration experiments reported in the manuscript, we repeated for at least four times. In addition, for each ligand to K63-Ub_2_ titration, we performed ligand to buffer titration accordingly, for separate background subtraction. The reported *K*_D_ values (as well as ΔH and ΔS) values are average ± SD over all the ITC fitting results.

Also, as the authors say that the mutations should have a stronger impact on the C1 state, it would therefore be informative that the authors do similar ITC measurements and analyses with a C1 ligand, e.g. the mentioned antibody, to further corroborate the conclusions.

We realized that the antibody-bound maybe not be an ideal ligand to illustrate the target recognition mechanism of K63-Ub_2_. Firstly, an antibody/antigen complex is much tighter than a typical signaling complex, and therefore induced fit may play a bigger role in the binding process. Secondly, in the antibody complex structure of K63-Ub_2_ ((PDB codes 3DVN and 3DVG), two extra residues were left at the N-terminus of the distal unit (Newton et al., Cell 2008), which may perturb the conformational space of K63-Ub_2_. We also predicted the inter-subunit PREs with MTS conjugation at K48C site, had the ligand-free K63-Ub_2_ adopts a structure similar to this complex structure. The resulting PRE profile is different from the experimental one, and therefore, such structure may not be present at a significant population for ligand-free K63-Ub_2_.

During the revision process, we found another physiological ligand of K63-Ub_2_, the fourth zinc finger domain of ubiquitin editing enzyme A20. Though A20 ZnF4 specifically binds to Lys63-linked polyubiquitin (Bosanac et al. Mol. Cell. 2010), the structure (PDB code 3OJ3) was only determined for the complex of A20/ubiquitin monomer (which however contains multiple ubiquitins in one asymmetric unit). This was the reason that we overlooked this ligand in the first place. The C1-state structure obtained for ligand-free K63-Ub_2_ is very similar to the complex structure with ZnF4. When introducing E64R_P_ mutation, the binding affinity between K63-Ub_2_ and ZnF4 is weakened by about three fold.

4) The authors should illustrate that the spin labels introduced do not directly interfere with either the open or the closed state structures, or potentially increase the tendency towards a closed state structure. This should be shown by highlighting the position of K11/K48 in structural figures (e.g. Figure 3) for K63-Ub_2_ alone, and when to bound to the two ligands investigated.

In addition, the authors should also verify that the spin labels themselves do not affect binding of the ligands, i.e. by comparing ITC data for wild type and spin labeled proteins.

We have illustrated the mutation site and conjugated paramagnetic probe in the complex structures, shown in Figure 1—figure supplement 2 of the revised manuscript. These modifications are distant from the bound ligands and from the other subunit.

We have compared paramagnetic spectrum with the reference spectrum, and we assessed the binding affinities for the modified K63-Ub_2_ using ITC towards NZF and tUIM. Among the two conjugation sites we used previously, K11C conjugated protein displays some chemical shift perturbations, and the bindings affinity towards tUIM is significantly increased. Finally, we identified N25C site at the distal unit that gives good inter-subunit PREs: conjugation of the paramagnetic probe at this site does not cause NMR chemical shift perturbations, nor affect the binding affinities towards K63-Ub_2_ ligands. In addition, conjugation of a different kind of paramagnetic probe, EDTA-Mn^2+^, at N25C site, affords inter-subunit PREs of similar profile as the MTS probe. These control experiments are presented in Figure 1—figure supplement 3, Figure 1—figure supplement 4 and Figure 1—figure supplement 6 of the revised manuscript.

5) The authors propose that the reduced τ_c_ values for the charge reversal mutation indicates a shift to more open conformations. However, it could also be that the charges also reduce the (known) tendency of (di)ubiquitin to dimerize. Can the authors rule out this possibility? Have they measured relaxation data for the monomer harboring the same charge reversals?

We performed the R_1_/R_2_ experiments at a lower protein concentration (50 µM). The difference in τ_c_ values of the wild type and mutant K63-Ub_2_ diminishes. Therefore, it is likely that the difference we observed at higher concentration (500 µM) is due to different dimerizing tendencies for the wild type and mutant proteins. Thus we removed related sections about relaxation measurements.

6) The Abstract is phrased as addressing a biophysical rather than an important biological/biochemical problem. Also the compact introduction may appeal more to a smaller expert community. On both I would suggest to change the emphasis towards the biochemical result and its possible biological impact, and add a short section to the Introduction and Discussion.

We have carefully revised the Abstract, Introduction and final Discussion, in particular about how different conformational states are involved in different functions of K63-Ub_2_. We believe the manuscript is more appealing to the general readers of *eLife*.

7) What is the structural basis for the effect of the charge reversal mutant?

In the closed-state structures, residue Glu64 of the proximal unit is opposing the positively charged residues Arg72 and Arg74 of the distal unit. Judging from the closed-state structures (illustrated in Figure 4—figure supplement 1), the E64R mutation should have a larger impact on the C1 state than one the C2 state, which were supported by the experimental data. Note that electrostatic attraction or repulsion is 1/r distance dependent, and can be effective even though the C-terminal tail of the distal unit is flexible.

8) Published literature where multi-domain conformations have been studied should be appropriately referenced, i.e. work by the groups of Berlin/Fushman (theoretical analysis), Blackledge, Luchinat, Sattler, Grzesiek, Kay and others (using PRE, RDC, SAXS data).

We have added several relevant references about characterizing the dynamics of multi-domain/subunit proteins.

[Editors' note: further revisions were requested prior to acceptance, as described below.]

1) The authors suggest that the use of SAXS data, which could help fitting populations, is not feasible as K63-Ub_2_, or mono-Ub dimerize non-covalently and this strongly affects the scattering. This would however also jeopardize the overall conclusions as one wonders whether the PRE data might also reflect intermolecular PREs? It would be interesting to show the SAXS data attached to a primary figure and their concentration dependence to judge how strong these effects really are.

We have performed SAXS measurements for K63-Ub_2_ at different concentrations. The *D*_max_ values of the SAXS data are larger at higher concentrations than the values at lower concentrations. This indicates the contribution from intermolecular species, presumably high-order noncovalent ubiquitin oligomers. The SAXS data appear reasonably well at relative low concentrations; the data recorded for 500 µM K63-Ub_2_ sample are similar to that at 1 mM.

The experimental SAXS *P*(r) profile has a narrower distribution than the curves computed for all known K63-Ub_2_ structures in the open state, manifesting the presence of a significant population of the closed compact state. We linearly combined the SAXS data calculated for the closed and open states at 70% and 30% populations and without further refinement, and we were able to recapitulate the experimental data with a chi-square of 1.24. Nevertheless, owing to the limited resolution of SAXS and possible contribution from higher-order oligomers, multiple conformational states cannot be resolved from the SAXS.

The SAXS data are now reported in Figure 2 (including two supplementary figures) of the revised manuscript. The intermolecular PREs were subtracted as described in the manuscript.

*2) Concerning the ITC data: The authors should additionally report the stoichiometries fitted for the ITC data. For better comparison* K_*D*_
*not* K_*A*_
*values should be reported consistently*.

*K*_D_ values are now reported in the main text and in the figures.

The binding stoichiometries are reported in Figure 1—figure supplement 3 and in Figure 5.

3) The Introduction addresses a general audience now, the Abstract not really yet. It could still be improved:

In short (I have no objection what parts would be used..): The authors show that free K63-Ub_2_ exists as a dynamic ensemble consisting of multiple closed and extended quaternary arrangements. This quaternary structural plasticity of K63-Ub_2_ enables it to be recognised in a variety of signalling pathways. Upon binding to a target protein one of the quaternary arrangements is selected. The authors show that different target proteins or point mutations can select or shift the equilibrium between different quaternary arrangements.

We have rewritten the Abstract. We appreciate the constructive suggestions.

4) Possible error: In the subsection headed “NMR data collection”: [Disp-formula equ3] and [Disp-formula equ4] seem to be partially swapped (Is Γ2 in the dia term correct? If not, [Disp-formula equ7] might also be switched.)

We have corrected [Disp-formula equ3] and [Disp-formula equ4]. [Disp-formula equ7] is correct.